# Behaviours involved in the role of victim and aggressor in bullying: Relationship with physical fitness in adolescents

**Juan de Dios Benítez-Sillero**[1,2]☯*, **Diego Corredor-Corredor**[3]☯, **Rosario Ortega-Ruiz**[2,4]‡, **Francisco Córdoba-Alcaide**[2,4]‡

**1** Department of Specific Didactic, Universidad de Córdoba, Córdoba, Spain, **2** Laboratory for Studies on Coexistence and Prevention of Violence (LAECOVI), Cordoba, Spain, **3** Counseling of Education, Junta de Andalucía, Andalucía, Spain, **4** Department of Psychology, University of Córdoba, Cordoba, Spain

☯ These authors contributed equally to this work.
‡ These authors also contributed equally to this work.
* eo1besij@uco.es

**Data Availability Statement:** All relevant data are within the manuscript and its Supporting information files.

## Abstract

Physical fitness is related to well-being and health. Adolescence is a key period in the psychological and social development of the person, in which interpersonal relationships gain strength, being bullying a type of violence that can affect the personality of those involved. At present, there is not enough research to determine the relationships between bullying and physical condition. The purpose of this study is to find out if there are any relationships among physical fitness, victimisation, and aggression in bullying, and to identify these behaviours. This is a descriptive study done in 1035 adolescents aged 12 to 17 years (M = 14.67, SD = 1.49). The European Bullying Intervention Project Questionnaire (EBIPQ) scale was used, and anthropometric characteristics of weight and height were measured. In addition, physical fitness tests from the Eurofit battery, sit-and-reach, 30-second sit-ups, horizontal jump, manual dynamometry, and 20-meter Multistage Shuttle Run Test (SRT) were included. The relationships between variables were analysed using Spearman correlations, linear regressions, and ordinal regressions. The most relevant findings indicate an inverse relationship between being a victim of bullying and having a better cardiorespiratory cardiovascular endurance. These also show a direct relationship between being a bully and skeletal muscle strength measured through the horizontal jump, 30-second sit-ups, and manual dynamometry tests. Theoretically, we can conclude that physical condition can be considered a predictor to consider in bullying. Specifically, cardiorespiratory fitness, in addition to its multiple physical and mental benefits, may be a protective element against bullying victimisation. In contrast, muscular strength, especially in boys, may be an important predictor, especially in the physical component, of aggression in bullying.

**Funding:** This project was funded by the Directorate General for Innovation and Teacher Training of the Counseling of Education of Andalusia (PIV-021/20) as an educational research project to be implemented in 2020. The funders had no role in study design, data collection and analysis, decision to publish, or preparation of the manuscript.

**Competing interests:** The authors declare that they have no conflicts of interest.

## Introduction

The wellbeing and health of individuals starts with the interest that society shows in the health and wellbeing of the child population. From its inception in 1988, the Eurofit test battery has become the most popular one used to assess the physical fitness of European children and adolescents, and the effectiveness of national physical education curricula [1,2]. The physical fitness construct is related to health and includes a number of elements, such as, in general terms, aerobic capacity, strength, muscular endurance and flexibility as well as body composition [3].

In children and adolescents, favourable associations have been reported linking cardiorespiratory and musculoskeletal fitness with lower risk of cardiometabolic diseases, adiposity, as well as musculoskeletal fitness with bone health [4,5] or with lower oxidative stress according to aerobic endurance [6].

Likewise, physical fitness is also related to psychological well-being and health, influencing, among others, social competence [7], mental health and cognition [4,5]. Aerobic capacity is associated with happiness and life satisfaction in Spanish adolescents [8] as well as preventing the negative health consequences of adolescent stress [9], improving social health [10] or enhancing resilience [11], although these psychological effects are more unknown than the physical ones [8]. However, musculoskeletal strength may be associated with aggression, especially in boys [12,13].

Adolescence is a critical and vulnerable period in several areas of physical, psychological and social development [14]. Some of these areas are of particular importance, given the changes that occur from puberty to late adolescence. Physical activity, exercise and fitness are likely to play a crucial role in helping the adolescent brain to develop [15].

In these years in which social activity is intensifying, and interpersonal relationships are becoming more and more important in the lives of adolescents, phenomena such as bullying and cyberbullying are acquiring a presence that, without being massive, affects a significant number of boys and girls. Bullying is a phenomenon of intentional interpersonal aggression between schoolchildren, repeated and sustained over time, in which there is an imbalance of power and dominance between the aggressor and the aggressed [16–18]. Most of these behaviours take place at school, especially at times and places where there is less supervision [19,20]. Both bullying and cyberbullying are dynamic and complex phenomena in which the personality of each individual brings important nuances. Many schoolchildren who have participated, either as victims or as aggressors of their peers, manage to get out of these dynamics in a short time, overcoming the pernicious effects that prolonging these problems over time would have on them, but others do not. Others maintain the situation for a prolonged period of time, leading to a certain chronification of their social problems that prolongs the stress and discomfort and possibly ends up affecting their physical, psychological and social health [21]. Undoubtedly, in addition to personal factors, these problems are influenced by contextual elements, particularly the psychosocial systems that organise school coexistence. It is therefore important to stop bullying and cyberbullying in time if the aim is to achieve quality in terms of health and well-being [17,18,21].

Victims of bullying may be subjected by one or more bullies and feel that they have fewer resources to get out of these situations and more difficulty defending themselves [22,23]. Bullies appear to have problems related to emotional regulation [24,25]. They do not usually show social skills deficits and enjoy popularity among their peers, possibly because their aggressive and overbearing behaviour patterns help them to gain or maintain that social status [26].

Some of the most studied aspects in adolescents that could be more linked to physical condition are variables, such as age, weight, height and gender in relation to bullying. In terms of

age, the time of highest incidence of bullying behaviour among schoolchildren is between 11 and 15 years [27,28] data generally indicating a curvilinear trend in early adolescence [29–31]. About gender, boys are usually more involved than girls [29–31]. Weight and height have been studied in the past associated with bullying. It has been seen a relationship between them although usually the variable body mass index has been used, and relationships have been found with obese and overweight subjects, with normal weight subjects being less studied [32,33].

Few studies have analysed the relationship between bullying and physical fitness in adolescents [34,35] these focusing on the role of the victim and concluding that high physical fitness may be a protective factor against bullying victimisation. García-Hermoso et al [34] studied 7,714 adolescents (9–17 years), and found that boys (50.27 ± 3.75) and girls (35.85 ± 3.05) not victimized in traditional bullying presented higher levels of cardiorespiratory fitness than boys (49.73 ± 3.79; $p = < .001$) and girls (35.50 ± 2.96; $p = .037$) victimized. While there were no differences in muscle strength measured with the manual dynamometry test. Hormazabal-Aguayo et al [36], related a healthy level of cardiorespiratory fitness to lower levels of bullying involvement in the role of victim, while the work of Greenleaf et al [35] focused on the teasing and verbal harassment adolescents receive because they are overweight. These authors studied 1,419 adolescents (12.41 ±.97 years), they found differences in the progressive aerobic cardiovascular endurance run (PACER) test, obtaining results for not teased (34.93 ± 17.03) and the teased (25.98 ± 13.54; F = 7.07; $p = .008$), they also found differences in the Push-ups test, not teased (16.60 ± 9.03) and the teased (12.30 ± 3.98; F = 5.69; $p = .017$), while no differences were found in the curl-ups and sit-and-reach tests.

In pre-adolescents, an inverse relationship has also been found for victimisation in peer relationships, especially in cardiorespiratory fitness, without being as conclusive with strength, nor in relation to aggression [10,37]. Fernandez-Bustos et al [10] found negative relationships between the tests of cardiorespiratory fitness (r = -.257), handgrip strength (r = -.118) and standing broad jump (r = -.302) with victimization in peer relationships, on the other hand, they found no relationships between the tests of cardiorespiratory fitness, handgrip strength and standing broad jump with aggressiveness in peer relationships. Chen et al [37] found positive relationships between more positive peer relationships and cardiorespiratory fitness (r = .18), muscular strength (r = .27) and muscular endurance (r = .27) in 550 subjects (11.84 ±.52 years). Therefore, the relationships between physical fitness and bullying in adolescents are rather unexplored, especially linked to aggressive behaviour.

Therefore, the present study is based on the hypothesis that physical fitness may be related to the incidence of bullying, being a preventive factor for victimisation and still unknown for aggression. Two main objectives are proposed, the first one is to know the relationships between physical fitness and victimisation and aggression in bullying; and the second one is to know which behaviours associated with aggression and victimisation in bullying are more specifically related to cardio-vascular endurance, muscular strength and flexibility as components of physical fitness.

## Materials and methods

### Design

A descriptive, exploratory, cross-sectional, descriptive study was carried out.

### Participants

The study involved 1035 schoolchildren aged 12 to 17 years ($M$ = 14.67, $SD$ = 1.49), of whom 506 were girls (48.9%), from four public schools in southern Spain (three in the province of

Cordoba and one in Huelva). The participants belonged to the first to fourth year of secondary education (12 to 16 years) and first year of baccalaureate (17 years), if they had not repeated a year. Subjects above that age were discarded. Participants who showed any injury or pathology to perform the physical tests were excluded from the study.

## Procedure

The present study was carried out after obtaining the respective permits from the school councils of the participants' schools, as well as signed informed consent forms from their families. The project was also approved by the University of Cordoba Human Research Ethics Committee. The children were told about the main aim of the study and that their participation would be anonymous, confidential, and voluntary. The computer-based questionnaires were administered in the classroom. The average time for completing the questionnaire ranged from 20 to 30 minutes. The data were collected from November 2018 to December 2020.

## Data analysis

The preliminary analyses were carried out using mean and standard deviation. They compare the scores between the values of our study with the percentiles of European [1] and national [41] reference studies. Spearman correlations were calculated between the study variables. Linear regression analysis and stepwise regression analysis were performed for victimisation and for aggression as dependent variables, including variables such as age, gender, BMI, strength and cardiorespiratory fitness as predictors. For the analysis of the influence of behaviours derived from the questionnaire questions, an ordinal regression model was conducted. For the linear and ordinal regression analysis models, the flexibility variable was discarded as it had no relationship in the previous analysis. The muscle strength variables were grouped together using the same z-score, calculating the mean of the sum of the 3 variables of horizontal jump, best hand manual dynamometry and abdominal sit-up in 30 seconds. Given the cross-sectional nature of this study, this prediction can only be interpreted on a theoretical basis and no causal relationship can be established.

## Instruments

**Victimisation.** To measure the incidence of bullying, we used the victimisation scale of the European Bullying Intervention Project Questionnaire (EBIPQ) instrument (Spanish version found in [18], consisting of 7 items with Likert-type response options from 0 to 4, where 0 = never, 1 = once or twice, 2 = once or twice a month, 3 = about once a week, and 4 = more than once a week. In this study, we focused exclusively on victimisation. The internal consistency values of the test were optimal: α = .85. These behaviours are described in Table 1.

**Aggression.** To measure the incidence of bullying, the victimisation scale of the European Bullying Intervention Project Questionnaire (EBIPQ) [18], composed of 7 items with Likert-type response options from 0 to 4, where 0 = never, 1 = once or twice, 2 = once or twice a month, 3 = about once a week and 4 = more than once a week, was used. In this study, aggression was considered exclusively. The internal consistency values of the test were optimal, α = .79. These behaviours are described in Table 1.

**Anthropometric characteristics.** All measurements were made with barefoot subjects and light clothing, dedicating the first day solely to the anthropo-metric data collection. The weight in kilograms (kg) and the percentage of fat was measured with a Tanita BF 350 scale (precision 0.1 kg). A SECA stadiometer (precision 0.1 cm) was used for measuring the high of the students. Based on these data, the body mass index (BMI = weight (kg)/height (m$^2$)) was calculated.

**Table 1. The questions related to bullying behaviours were the following (EBIPQ Ortega-Ruiz et al. [18].**

| |
|---|
| *Victimisation* |
| 1. I have been hit, kicked or pushed by someone. |
| 2. Someone has insulted me. |
| 3. Someone has said offensive words about me to other people. |
| 4. Someone has threatened me. |
| 5. Someone has stolen or broken my things. |
| 6. I have been excluded, isolated or ignored by others. |
| 7. Someone has spread rumours about me. |
| *Aggression* |
| 8. I have hit, kicked or pushed someone. |
| 9. I have insulted and said offensive words to someone. |
| 10. I have said offensive words about someone to other people. |
| 11. I have threatened someone. |
| 12. I have stolen or damaged something from someone else. |
| 13. I have excluded, isolated, or ignored someone. |
| 14. I have spread rumours about someone. |

**Fitness tests.** Before starting the measurements, a familiarization session was held in order to guarantee the standardization, validation and reliability of the measurements. The physical fitness tests were carried out during the students' physical education classes, between 9:00 and 13:00, depending on the timetable of the different classes. Five integrated tests were carried out within the EUROFIT battery [38]. The scientific reason for the use of these tests in adolescents has been previously published [39]and used in the international studies AVENA and HELENA [4]. All students are trained in these tests and are regular used as a part of their assessment tools throughout their education. The battery includes the tests described below:

a) **30 seconds sit-up tests**. It is a trunk power test in which the subject tries to execute as many sit-ups as possible, during a period of 30 seconds. The subject lies on his back on a mat, places his hands behind the back of his neck and his legs are flexed 90 degrees with his feet supported, then the subject must be incorporated until he touches the knees with his elbows. The number of correct executions of the movement will be noted. For its development we used a mat for each subject and a CasioHS-80TW chronometer.

b) **Sit-and-reach flexibility test**. This test measures the range of motion of the hip. With the subject seated on a mat, with the legs extended and resting the soles of the feet on a standardised box, the subject must flex the trunk forward, trying to reach as far as possible with the outstretched hands, moving with the tips of the fingers a ruler located on the surface of the box, which has a graduation in centimetres. According to the protocol, the measure of 15 centimetres is placed in the situation where the feet rest.

c) **Horizontal jump**. It was used to assess lower body explosive muscular strength. The subject starts from a static position, located immediately after a line, with feet shoulder-width apart and parallel, having to make a jump as far as possible without losing balance in the fall. Two attempts were made on a hard, non-slip surface, scoring the best of them. The result was recorded in centimetres, using a tape measure.

d) **Handgrip strength**. The manual grip strength in kilograms was assessed in both hands, using a TAKEY TKK 5110 dynamometer (interval 5–100 kg, precision 0.1 kg), with adjustable handle. The subject takes the dynamometer with the hand and keeping it slightly away

from the body. The subjects must press gradually and continuously for 2 seconds. The optimal grip was calculated by the equation of Ruiz et al. [40]. Two attempts were made with each hand, scoring the best result. For this study, the most common measure of best performance with one hand was used [38] although for comparison with the Spanish reference data [41] the measure of the sum of both hands was used.

e) **20 meters shuttle run test**. This is a maximal incremental field test that evaluates the maximum aerobic capacity indirectly. The subject performs a roundtrip race in 20 meters. The test begins with a slow pace. They should make the changes of direction at the moment of the sound signal that progressively accelerates. The test ends when the subject is not able to follow the rhythm that is imposed. The time in minutes and seconds was determined to increase the precision of the measurements. The author's original software in mp3 version [42].

To avoid alteration between tests, tests were performed on different days in the following order: day two (sit-and-reach flexibility test and sit-up tests), day three (20-m shuttle run), day four (handgrip and horizontal jump) [43].

## Results

Tables 2 and 3 present the descriptive data of the physical fitness tests according to age in boys and girls in relation to the percentiles of the reference data for the European and Spanish population. The physical fitness values in boys were close to the 50th percentile, with some lower values especially in the flexibility and manual dynamometry tests. While in girls, the physical fitness values were close to the 50th percentile, with the lowest results in the flexibility and manual dynamometry tests.

Table 4 presents the results concerning victimisation and aggression. Victimisation correlates positively with age and weight, and inversely with the horizontal jump and the SRT. Aggression correlates positively with height, weight, the 30-second sit-up test, the horizontal jump and manual dynamometry.

In the linear regression model proposed in Table 5, the results are presented for victimisation as the dependent variable, with age, sex, BMI, cardiorespiratory fitness, and musculoskeletal strength as predictors. The regression models were statistically significant in victimisation

**Table 2. Descriptive data of age-specific physical fitness test results in boys with comparative percentile based on baseline data.**

| Age | 12 | | | 13 | | | 14 | | | 15 | | | 16 | | | 17 | | |
|---|---|---|---|---|---|---|---|---|---|---|---|---|---|---|---|---|---|---|
| Test | M (SD) n = 95 | E (P) | S (P) | M (SD) n = 76 | E (P) | S (P) | M (SD) n = 103 | E (P) | S (P) | M (SD) n = 123 | E (P) | S (P) | M (SD) n = 96 | E (P) | S (P) | M (SD) n = 36 | E (P) | S (P) |
| Flexibility (cm) | 11.5 (6.80) | 20 | | 14.0 (6.38) | 30 | 40 | 12.9 (8.40) | 20 | 20 | 14.6 (8.70) | 20 | 20 | 15.3 (8.61) | 20 | 20 | 15.1 (7.96) | 10 | 10 |
| Sit-up (repetitions) | 21.0 (5.48) | 50 | | 24.0 (6.05) | 70 | | 24.6 (6.10) | 60 | | 28.2 (6.19) | 80 | | 26.4 (5.82) | 60 | | 28.0 (5.15) | 70 | |
| Horizontal jump (cm) | 161.2 (26.33) | 50 | | 167.1 (32.66) | 40 | 40 | 184.5 (29.89) | 50 | 60 | 186.0 (33.30) | 40 | 30 | 198.5 (35.65) | 40 | 40 | 203.1 (29.87) | 40 | 40 |
| Handgrip both hands (Kg) | 38.9 (9.60) | | | 44.1 (11.74) | | 20 | 57.6 (13.91) | | 30 | 62.1 (13.12) | | 20 | 67.1 (14.61) | | 20 | 70.8 (12.96) | | 50 |
| Handgrip best hand (Kg) | 20.6 (5.08) | 30 | | 23.1 (6.39) | 20 | | 30.3 (7.18) | 20 | | 32.7 (6.79) | 20 | | 35.1 (7.17) | 10 | | 37.1 (6.81) | 10 | |
| SRT (periods) | 5.1 (2.14) | 40 | | 5.3 (2.27) | 30 | 40 | 6.2 (2.24) | 40 | 40 | 7.3 (2.48) | 50 | 60 | 7.3 (2.61) | 50 | 40 | 7.8 (2.77) | 50 | 60 |

cm: Centimeters; Kg: Kilograms; SRT: 20-meter multistage shuttle run test; M: Mean; SD: Standard deviation; E: European references (Tomkinson et al 2018); S: Spanish references (Ortega et al 2005); P: Percentile. * The 20th percentile means that the result falls between the 20th and 30th percentile of the reference tables. There are no references for the Spanish population over the 12 years in this study.

**Table 3. Descriptive data of physical fitness test results by age in girls with comparative percentile based on baseline data.**

| Age | 12 | | | 13 | | | 14 | | | 15 | | | 16 | | | 17 | | |
|---|---|---|---|---|---|---|---|---|---|---|---|---|---|---|---|---|---|---|
| Test | M (SD) n = 108 | E (P) | S (P) | M (SD) n = 85 | E (P) | S (P) | M (SD) n = 80 | E (P) | S (P) | M (SD) n = 138 | E (P) | S (P) | M (SD) n = 89 | E (P) | S (P) | M (SD) n = 6 | E (P) | S (P) |
| Flexibility (cm) | 17.0 (8.63) | 20 | | 19.6 (7.89) | 30 | 30 | 19.1 (7.56) | 20 | 20 | 20.6 (9.71) | 20 | 20 | 21.8 (8.29) | 20 | 20 | 21.50 (13.07) | 20 | 30 |
| Sit-up (repetitions) | 18.7 (4.76) | 40 | | 20.3 (4.61) | 60 | | 21.0 (6.03) | 70 | | 23.4 (4.44) | 80 | | 20.5 (4.53) | 50 | | 17.0 (4.15) | 20 | |
| Horizontal jump (cm) | 142.0 (19.49) | 40 | | 136.4 (28.47) | 20 | 30 | 134.4 (27.78) | 20 | 30 | 145.1 (26.93) | 30 | 40 | 143.4 (27.40) | 30 | 30 | 137.0 (24.72) | 10 | 30 |
| Handgrip both hands (Kg) | 39.1 (8.51) | | | 38.2 (7.83) | | 10 | 44.0 (10.21) | | 20 | 45.1 (9.46) | | 20 | 45.1 (8.46) | | 20 | 36.7 (6.76) | | 0 |
| Handgrip best hand (Kg) | 20.8 (4.51) | 50 | | 20.3 (4.20) | 20 | | 23.2 (5.46) | 20 | | 23.7 (4.89) | 20 | | 23.9 (4.37) | 20 | | 20.0 (4.25) | 5 | |
| SRT (periods) | 3.5 (1.44) | 30 | | 3.7 (1.49) | 40 | 70 | 3.9 (1.81) | 40 | 70 | 4.6 (1.70) | 60 | 70 | 4.0 (1.59) | 40 | 40 | 2.5 (0.63) | 10 | 20 |

cm: Centimeters; Kg: Kilograms; SRT: 20-meter multistage shuttle run test; M: Mean; SD: Standard deviation; E: European references (Tomkinson et al 2018); S: Spanish references (Ortega et al 2005); P: Percentile. * The 20th percentile means that the result falls between the 20th and 30th percentile of the reference tables. There are no references for the Spanish population over the 12 years in this study.

analysis ($p$ = .002; $R^2$ = .018) and in aggression analysis ($p$ = < .001; $R^2$ = .038). The only variable that was inversely significant was cardiorespiratory fitness. In the aggression model as the dependent variable, the significant predictor variables were sex, being higher in boys, age, BMI and strength with a positive relationship.

Table 6 shows the stepwise multiple regression model. For victimisation only the SRT test is significant in the first step explaining 3.1% of victimisation. The aggression model was significant in the first step with strength explaining 2.6% of the aggression and in the second step with BMI, with an increased R2 of 0.6%.

If we analyze the stepwise multiple regression, entering all physical tests of strength independently. In the victimisation model, only the SRT test remains significant in the first step ($p$ = < .001; β = -.028; $R^2$ = .012). In the aggression model, there are four significant steps, in the first step the sit-up test like predictor ($p$ = < .001; β = .010; $R^2$ = .022), in the second step, sit-ups ($p$ = < .001; β = .007) and male sex ($p$ = .005; β = .074; $\Delta R^2$ = .008), in the third step, sit-ups ($p$ = .008; β = .006), male sex ($p$ = .003; β = .077) and age ($p$ = .005; β = .024; $\Delta R^2$ = .007) and in

**Table 4. Linear correlations with victimisation and aggression of anthropometric and physical fitness variables.**

| | Victimisation | | | Aggression | | |
|---|---|---|---|---|---|---|
| | r | inf. limit | sup. limit | r | inf. limit | sup. limit |
| Age (years) | .075* | .01 | .14 | .135*** | .07 | .19 |
| BMI (Kg/m$^2$) | .084** | .02 | .14 | .089** | .03 | .15 |
| Flexibility (cm) | -.010 | -.07 | .05 | -.020 | -.08 | .04 |
| Sit-up (repetitions) | .019 | -.04 | .08 | .125*** | .06 | .18 |
| Horizontal jump (cm) | -.063* | -.12 | .00 | .065* | .00 | .13 |
| Handgrip best hand (Kg) | ,001 | -.06 | .06 | ,108** | .05 | .17 |
| SRT (periods) | -.085** | -.15 | -.02 | .027 | -.03 | .09 |

* $p$ < .05;

** $p$ < .01;

*** $p$ < .001.

BMI: Body mass index; Kg: Kilograms; cm: Centimetres; SRT: 20-meter multistage shuttle run test; r: Spearman correlation; inf. Limit; inferior limit of the confidence interval; sup. limit; superior limit of the confidence interval.

**Table 5. Linear regression analysis with gender, age, BMI, strength and cardiorespiratory fitness as cross-sectional predictors of victimisation and aggression.**

|  | Victimisation | | Aggression | |
|---|---|---|---|---|
|  | β | t | β | t |
| Sex male | .039 | 1.021 | .084 | 2.201* |
| Age | .052 | 1.457 | .067 | 1.890 |
| BMI | .043 | 1.312 | .065 | 1.986* |
| Strength | -.008 | -.167 | .093 | 2.032* |
| SRT | -.123 | -3.023** | -.007 | -.166 |

\* p < .05;

\*\* p < .01

Standardised coefficients. BMI: Body mass index; SRT: 20-meter multistage shuttle run test.

**Table 6. Stepwise regression analysis with gender, age, BMI, strength and cardiorespiratory fitness as cross-sectional predictors of victimisation and aggression.**

| Victimisation | | | Aggression | | |
|---|---|---|---|---|---|
|  | β | R² |  | β | R² |
| *Step 1* | | | *Step 1* | | |
|  | -.031** | .018 | Strength | .078*** | .025 |
|  | | | *Step 2* | | ΔR² |
|  | | | Strength | .081*** | |
|  | | | BMI | .007* | .006 |

\* p < .05;

\*\* p < .01;

\*\*\* p < .001

Standardised coefficients. BMI: Body mass index.

the fourth with the predictors, step sit-up ($p$ = .004;β = .007), male sex ($p$ = .003; β = .078), age ($p$ = .022;β = .020;ΔR² = .007) and BMI ($p$ = .026;β = .007;ΔR² = .004).

Table 7 shows the ordinal regression model for the victimisation questions. The results for the 4 questions with the most significant results (3,4,6 and 7) are presented in the table. In addition, questions 1 and 5, which presented significant results, are described in text. In the model for question 1, the predictors of sex (ES = .632;$p$ = .001) were noticeable, being higher in boys and BMI (ES = .040;$p$ = .035). In question 5, only sex was important in the model (ES = .586;$p$=, 006) being higher in boys. Question 2 did not show any significant relationship.

Table 8 shows the ordinal regression model for the victimisation questions. The results for the 3 questions with the most significant results (8,9 and 12) are presented in the table. In addition, questions 10 and 11, which presented significant results, are described in the text. Question 10 shows a significant relationship in the age predictor model (ES = .234;$p$ = .000); question 11 in the sex model (ES = 1.414;$p$ = .000) being higher in boys. In Questions 13 and 14 none of the predictors was significant.

## Discussion

Few studies have analysed the relationship between physical fitness, including aerobic capacity, strength, muscular endurance and flexibility as well as body composition [3], and bullying in

**Table 7. Ordinal regression analysis with age, height, weight, Strength Cardiorespiratory fitness and gender, as cross-sectional predictors of victimisation behaviours.**

| | 3. Someone has said offensive words about me to other people. | | | | | 4. Someone has threatened me. | | | | | 6. I have been excluded, isolated or ignored by others. | | | | | 7. Someone has spread rumours about me. | | | | |
|---|---|---|---|---|---|---|---|---|---|---|---|---|---|---|---|---|---|---|---|---|
| | β | ES | W | p | IC 95% | β | ES | W | p | IC 95% | β | ES | W | p | IC 95% | β | ES | | p | IC 95% |
| Never | 2.67 | .76 | 12.2 | **.000** | 1.17–4.16 | .96 | 1.08 | .79 | .375 | -1.16–3.07 | 2.94 | .93 | 9,99 | **,002** | 1,11–4,75 | 3,52 | ,82 | 18,34 | **,000** | 1,91–5,13 |
| 1 or 2 | 4.19 | .77 | 29.6 | **.000** | 2.68–5.70 | 2.53 | 1.09 | 5.41 | **.020** | .40–4.66 | 4.36 | ,94 | 21,69 | **,000** | 2,53–6,20 | 5,17 | ,83 | 38,62 | **,000** | 3,54–6,80 |
| 1 or 2 month | 4.81 | .77 | 38.6 | **.000** | 3.29–6.33 | 2.95 | 1.09 | 7.28 | **.007** | .81–5.09 | 4.79 | ,94 | 25,94 | **,000** | 2,95–6,64 | 5,78 | ,84 | 47,61 | **,000** | 4,14–7,42 |
| 1 a week | 5.36 | .78 | 46.4 | **.000** | 3.79–6.85 | 3.36 | 1.10 | 9.32 | **.002** | 1.20–5.52 | 5.33 | ,95 | 31,53 | **,000** | 3,47–7,20 | 6,15 | ,84 | 53,30 | **,000** | 4,50–7,80 |
| Age | .16 | .5 | 10.8 | **.001** | .06 -.25 | -.03 | .07 | .18 | .668 | -.16 -.11 | .15 | ,06 | 6,37 | **,012** | ,03 -,26 | ,20 | ,05 | 15,21 | **,000** | ,10 -,30 |
| BMI | .02 | .02 | 1.26 | .262 | -.01 -.05 | -.00 | .02 | .03 | .854 | -.05 -.04 | ,01 | ,02 | ,070 | ,792 | -,03 -,04 | ,02 | ,02 | 1,72 | ,190 | -,01 -,06 |
| Strenght | .03 | .12 | .08 | .773 | -.19 -.26 | .13 | .15 | .68 | .408 | -.17 -.43 | -.19 | ,14 | 1,80 | ,180 | -,46 -,09 | ,24 | ,12 | 3,64 | ,056 | -,01 -,48 |
| SRT | -.08 | .03 | 5.58 | **.018** | -.15–-.01 | -.13 | .05 | 7.90 | **.005** | -.22–-.04 | -.16 | ,04 | 12,99 | **,000** | -,24–-,07 | -,09 | ,04 | 5,64 | **,018** | -,16–-,02 |
| Sex Male | -.27 | .16 | 2.80 | .094 | -.57 -.05 | .57 | .22 | 6.73 | **.010** | .14–1.00 | ,02 | ,19 | ,01 | ,914 | -,35 -,39 | -,70 | ,18 | 16,10 | **,000** | -1,1–-,36 |
| Nagelkerke R² | 0.34 | | | | | 0.21 | | | | | 0.43 | | | | | 0,65 | | | | |
| χ2 (gl) | 30.99 (5) *** | | | | | 14.28 (5) * | | | | | 34.48 (5) *** | | | | | 57.12 (5) *** | | | | |

* p < .05;

*** p < .001;

The reference category was more than once a week and the reference category in sex was male. W:Wald; BMI: Body mass index; SRT: 20- meter multistage Shuttle Run Test

adolescents [34,35]. This is why the present study set out to analyse the relationships between these variables. In general terms, an inverse relationship was found between being a victim of bullying and a better cardiorespiratory fitness measured through the SRT, and a direct relationship between being a bully and skeletal muscle strength measured through the horizontal jump, 30-second sit-ups and manual dynamometry tests.

**Table 8. Ordinal regression analysis with age, height, weight, Strength Cardiorespiratory fitness and gender, as cross-sectional predictors of aggression behaviours.**

| | 8. I have hit, kicked or pushed someone. | | | | | 9. I have insulted and said offensive words to someone. | | | | | 12. I have stolen or damaged something from someone else. | | | | |
|---|---|---|---|---|---|---|---|---|---|---|---|---|---|---|---|
| | β | ES | Wald | p | IC 95% | β | ES | Wald | p | IC 95% | β | ES | Wald | p | IC 95% |
| Never | 1.69 | 1.03 | 2.69 | .101 | -.33–3.71 | 3.68 | .80 | 20.96 | **.000** | 2.1–5.30 | 4.20 | 1.42 | 8.73 | **.003** | 1.41–6.98 |
| 1 or 2 | 3.71 | 1.04 | 12.65 | **.000** | -1.67–5.75 | 5.45 | .81 | 44.92 | **.000** | 3.86–7.05 | 6.59 | 1.46 | 20.45 | **.000** | 3.74–9.49 |
| 1 or 2 month | 4.18 | 1.05 | 15.79 | **.000** | 2.12–6.24 | 6.06 | .82 | 54.65 | **.000** | 4.45–7.66 | 7.41 | 1.50 | 24.26 | **.000** | 4.46–10.36 |
| 1 a week | 4.65 | 1.07 | 19.02 | **.000** | 2.56–6.74 | 6.76 | .83 | 66.02 | **.000** | 5.13–8.39 | 7.70 | 1.53 | 25.26 | **.000** | 4.70–10.70 |
| Age | -.03 | .07 | .25 | .616 | -.16 -.10 | .12 | .05 | 5.78 | **.016** | .02 -.22 | .07 | .09 | .59 | .441 | -.11 -.25 |
| BMI | .01 | .02 | .40 | .527 | -.03 -.05 | .05 | .07 | 8.23 | **.004** | .02 -.08 | .03 | .03 | .97 | .326 | -.03 -.08 |
| Strenght | .32 | .14 | 5.01 | **.025** | .04 -.61 | .02 | .19 | .03 | .875 | -.21 -.25 | .45 | .19 | 5.41 | **.020** | .07 -.82 |
| SRT | -.05 | .04 | 1.27 | .260 | -.12 -.03 | -.01 | .03 | .03 | .858 | -.07 -.06 | -.07 | .05 | 1.60 | .207 | -.17 -.04 |
| Sex Male | .92 | .22 | 18.15 | **.000** | .50–1.34 | .45 | .17 | 7.33 | **.007** | .12 -.77 | .72 | .31 | 5.48 | **.019** | .12–1.32 |
| Nagelkerke R² | 0.66 | | | | | 0.36 | | | | | 0.60 | | | | |
| χ2 (gl) | 47.92 | 5 | *** | | | 31.95 | 5 | *** | | | 30.24 | 5 | *** | | |

*** p < .001;

The reference category was more than once a week and the reference category in sex was male. W: Wald; BMI: Body mass index; SRT: 20- meter multistage Shuttle Run Test.

The study sample showed average levels of physical fitness compared to benchmark studies in European adolescents [1]and in Spanish adolescents [41]. Physical fitness in adolescence has been shown to be related to physical health [4], well-being and social competence [7].

In the present study, boys were found to be more aggressive, with no differences in victimisation, when both variables were controlled for the predictors of the model. The results are partially in line with previous research, as several studies indicate that involvement in aggression and victimisation is higher in boys than in girls, both in the case of victimisation and aggression [29–31]. For example, in the study by Cerezo et al in the Spanish population, 87% of the aggressors were male and 63.8% of the victims in a sample of 847 subjects aged 9–18 years [30].

In terms of age, higher results are usually found between 11 and 15 years of age [27–29,44]. However, in the present study, aggression shows a positive relationship with age when controlling for the predictors of the model, but not victimisation, which correlates positively when not controlled for other predictors, and the relationship between victimisation and age may be partly explained by other predictors of the model.

Considering the BMI, in the present study, it was positively related as it was not influenced by other predictors. However, this association disappeared in victimisation when the other predictors were introduced into the model and was maintained in aggression. This could be understood as meaning that the relationship between victimisation and higher BMI could be explained by other predictors influencing the model, in this case, cardiorespiratory endurance, while the relationship between BMI and aggression continues to maintain its positive relationship. Therefore, the positive relationships found in normal weight [33] and obese and overweight [32] subjects are evident in our study when other predictors of victimisation, such as cardiorespiratory fitness, are not taken into account in the victimisation model. So far, there are studies that have found a relationship between BMI and aggressiveness [12,45–47] although few studies have analysed the role of aggressiveness in bullying. Janssen et al [48] found a relationship with overweight and obesity only in 15–16 year olds and not in other adolescent ages, although the relationship between BMI and aggression in normal weight boys is not determined.

In the relationships between physical fitness tests and victimisation and aggression, no significant correlations were found in the case of the flexibility test in adolescents, as in the study by Greenleaf et al [35] which analyses victimisation through teasing. A positive and significant correlation was found with peer relationships in pre-adolescents (r = .16) [37]. In general, this test has not been used as much in the study bullying in the literature analysed, being its study more related to physical well-being, such as back and neck pain [49]. For this reason, after verifying the non-existence of correlations, it was not used in the regression models.

The present study seems to reinforce the relationship between victimisation and cardiorespiratory fitness as measured by the SRT. This was the only significant predictor in the linear regression model. Previous studies in adolescents agree with these results [34,35]. Greenleaf et al [35] found better cardiorespiratory fitness (p = .008;Cohen's d = 0.58) in adolescents (M = 12.41 years) not teased and Garcia-Hermoso et al [34] found higher levels of victimisation in traditional bullying among boys with worse cardiorespiratory fitness, both adjusted for age, pubertal status and excessive TV use and unadjusted for boys and girls, as well as a negative correlation between physical test and victimisation in traditional bullying (r = -11). In this sense, we consider the use of other predictors that are closely related to victimisation and cardiorespiratory fitness itself, such as age, sex and BMI. These are a strong point of the analysis of the present study, with significant inverse relationships between cardiorespiratory endurance and victimisation. In the strength tests, there was also not only a negative correlation with horizontal jumping and victimisation. Garcia-Hermoso et al [34] also found no relationship

with the manual dynamometry test. Greenleaf et al [35] found higher levels of push-ups in not teased subjects (p = 0.17), but not in the curl-up test.

In terms of specific victimisation behaviours, in relation to the questions in the questionnaire, the negative relationship of cardiorespiratory fitness with behaviours related to having suffered indirect, verbal and relational bullying, through offensive words to the victim, being excluded, isolated or ignored, or being defamed through rumours and others, such as receiving threats, stands out. Few studies have analysed these relationships as reflected in the review of the literature reviewed. Only Greenleaf et al. [35], refers to teasing behaviours, but specifically related to weight.

Although much remains to be explored about these relationships, there appears to be evidence that being verbally bullied may be associated with lower levels of self-esteem, depression, physical self-concept and self-efficacy for physical activity, which may in turn be linked to lower levels of physical fitness [34]. Other authors also attribute importance to how possessing greater cardiorespiratory fitness may promote greater resilience by strengthening self-regulation through top-down control of bottom-up processing [15].

In relation to aggressiveness in bullying and physical fitness tests, this was directly related to the three muscle strength tests, and to the strength variable resulting from combining the three once the model was influenced by the other predictors. If we consider the physical fitness tests in our study, apart from the strength component derived from analysing the three strength tests together, it seems to be the abdominal test that is most associated with aggressive behaviours in bullying. Of the studies that have analysed the relationship between physical fitness and bullying, only one [10] has analysed the aggressiveness component, but related to peer relationships and in pre-adolescents, no relationship was found between the physical tests of handgrip strength and standing broad jump and aggression in peer relationships. The present study yields new results in this sense, finding relationships with the manual dynamometry test, with higher levels of aggressiveness in subjects with higher levels of strength, and no relationship with the horizontal jump test. In any case, these results should be viewed with caution, as muscle strength experiences a significant increase in adolescence over pre-adolescence [50]. On the other hand, Chen et al [37] found a positive relationship between having best scores in the questionnaire of peer relations and the horizontal jump test (r = .18), although with this questionnaire used, it cannot be determined, that such peer relationships are related to the bullying components of victimisation and aggressiveness. Looking at specific behaviours, those related to musculoskeletal strength were linked to behaviours of physical violence, having hit, kicked or pushed someone, and those related to stealing or damaging something from someone, which were also higher in boys. There is previous evidence of the relationship between strength, a component of physical fitness, with aggressiveness although it has not been specifically studied in bullying, especially in boys as in the present study, with testosterone being one of the most relevant hormones among the possible causes of this relationship [12,13]. These findings would be in line with defining aspects of bullying for example, Olweus' classic definition [51]as an imbalance of power, which could be understood in this case as an imbalance of forces between the abuser and the abused, with physical strength in adolescents as a component of physical condition and as an element to be considered in this specific case.

## Conclusions

In short, the results of the present study lead us to the theoretical basis conclusion that physical fitness can be considered a predictor element to be considered in relation to bullying. Specifically, cardiorespiratory fitness, in addition to its multiple physical and mental benefits, may be a protective element against bullying victimisation. In contrast, muscular strength, especially

in boys, may be an important predictor, especially in the physical component, of aggression in bullying.

## Limitations of the study, proposals for future work, and educational implications

Among the limitations of the study are that the cross-sectional measurement limits the understanding of the relationships between the study variables. It would be interesting to test the relationship between participation in more and less physical activity, as children with better peer relationships and higher self-esteem participate more in moderate and intense physical activity, which influences better physical fitness [10]. It can also be argued that relatively early intervention in adolescent health-related risk factors may not only improve health and fitness, but also reduce bullying victimisation and delinquent incidents in the school context [52].

One of the main educational implications derived from the results of the present study has to do with the need of including aspects related to physical fitness in preventive programmes against bullying, with Physical Education being a potentially beneficial subject for this purpose [53,54]. There are already authors who show that anti-bullying programmes do not work specific contents on bullying linked to physical condition, specifically weight [55].

## Supporting information

**S1 File. Renamed_2ac5e.**
(XLSX)

## Author Contributions

**Conceptualization:** Juan de Dios Benítez-Sillero, Rosario Ortega-Ruiz, Francisco Córdoba-Alcaide.

**Data curation:** Juan de Dios Benítez-Sillero.

**Formal analysis:** Juan de Dios Benítez-Sillero, Rosario Ortega-Ruiz.

**Funding acquisition:** Juan de Dios Benítez-Sillero, Rosario Ortega-Ruiz, Francisco Córdoba-Alcaide.

**Investigation:** Juan de Dios Benítez-Sillero, Diego Corredor-Corredor, Rosario Ortega-Ruiz, Francisco Córdoba-Alcaide.

**Methodology:** Juan de Dios Benítez-Sillero, Rosario Ortega-Ruiz, Francisco Córdoba-Alcaide.

**Project administration:** Juan de Dios Benítez-Sillero, Francisco Córdoba-Alcaide.

**Resources:** Juan de Dios Benítez-Sillero.

**Software:** Juan de Dios Benítez-Sillero.

**Supervision:** Juan de Dios Benítez-Sillero, Rosario Ortega-Ruiz, Francisco Córdoba-Alcaide.

**Validation:** Juan de Dios Benítez-Sillero, Rosario Ortega-Ruiz.

**Visualization:** Juan de Dios Benítez-Sillero, Rosario Ortega-Ruiz, Francisco Córdoba-Alcaide.

**Writing – original draft:** Juan de Dios Benítez-Sillero, Diego Corredor-Corredor, Rosario Ortega-Ruiz, Francisco Córdoba-Alcaide.

**Writing – review & editing:** Juan de Dios Benítez-Sillero, Diego Corredor-Corredor, Rosario Ortega-Ruiz, Francisco Córdoba-Alcaide.

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
