## [Decision Letter · Decision Letter 0]

7 Jul 2021

PONE-D-21-09225

Behaviours involved in the role of victim and aggressor in bullying: relationship with physical fitness in adolescents.

PLOS ONE

Dear Dr. Benítez-Sillero,

Thank you for submitting your manuscript to PLOS ONE. After careful consideration, we feel that it has merit but does not fully meet PLOS ONE’s publication criteria as it currently stands. Therefore, we invite you to submit a revised version of the manuscript that addresses the points raised during the review process.

I reviewed your manuscript carfully  to determine suitability for publication based on a combination of factors, including whether the topic is well suited to the aims and scope of the journal, methodological considerations, and whether the findings make a sufficient contribution to the existing literature. Based on my proper review and external reviewers, your paper could be considered further for publication with major revisión.

One of ther reviewer explain some concerns that should be attended:

1- Complete two additional tests analysis: Multiple stepwise regression analysis, with the aim of identifying the most related component of physical condition with dependent variables.

One-way ANOVA to analyze whether there are differences in physical condition between subjects with differential points in victimisation and aggression in bullying

2- Rewrite their Introduction and Discussion. previous studies have analyzed the object of study of this manuscript, please consider it necessary to include quantitative results of these studies in the introduction section.

We look forward to receiving your revised manuscript.

Kind regards,

Celestino Rodríguez, PhD

Academic Editor

PLOS ONE

Journal Requirements:

2. Please improve statistical reporting and refer to p-values as "p<.001" instead of "p=.000". Our statistical reporting guidelines are available at https://journals.plos.org/plosone/s/submission-guidelines#loc-statistical-reporting.

“This project was funded by the Directorate General for Innovation and Teacher Training of the Counseling of Education of Andalusia (PIV-021/20) as an educational research project to be implemented in 2020.”

“This project was funded by the Directorate General for Innovation and Teacher Training of the Counseling of Education of Andalusia (PIV-021/20) as an educational research project to be implemented in 2020.”

4. We suggest you thoroughly copyedit your manuscript for language usage, spelling, and grammar. If you do not know anyone who can help you do this, you may wish to consider employing a professional scientific editing service.

Reviewers' comments:

Reviewer's Responses to Questions

**Comments to the Author**

1. Is the manuscript technically sound, and do the data support the conclusions?

Reviewer #1: Partly

Reviewer #2: Yes

2. Has the statistical analysis been performed appropriately and rigorously? 

Reviewer #1: No

Reviewer #2: Yes

3. Have the authors made all data underlying the findings in their manuscript fully available?

Reviewer #1: No

Reviewer #2: Yes

4. Is the manuscript presented in an intelligible fashion and written in standard English?

Reviewer #1: Yes

Reviewer #2: No

5. Review Comments to the Author

Reviewer #1: Dear Authors,

Priorities I congratulate the authors for their ideas and efforts. The time they spend working is very valuable. In the attached document I provide you with a series of comments about the manuscript.

Best regards.

Reviewer #2: The article is well supported theoretically and the references are current and related to the objective of the research.

As aspects of improvement, the following are recommended:

- It would be interesting to have more data regarding the sample and the data collection procedure. Has any participant had to be ruled out for medical reasons? Have you ever wondered if the participants have any pathology that could interfere with their performance?

- There is no data on how the fitness tests were carried out in the sample (if, for example, it was part of the physical education class or if the measurements were made in the same time slot)

- Finally, indicate that the writing of the manuscript should be revised to improve its reading: the theoretical introduction presents short sentences that can be joined (for example, line 77, line 250 R2 should be corrected)

6. PLOS authors have the option to publish the peer review history of their article (what does this mean?). If published, this will include your full peer review and any attached files.

Reviewer #1: No

Reviewer #2: No

---

## [Author Response · Author response to Decision Letter 0]

22 Sep 2021

Juan de Dios Benítez Sillero

Behaviours involved in the role of victim and aggressor in bullying: relationship with physical fitness in adolescents.

Dear Editor I enclose the letter of response to the reviewers, as well as some comments from the editor. For ease of interpretation, we have put the authors' comments in red. I hope I have solved the problems of the article and carried out a satisfactory process based on the reviewers' suggestions.

Reviewers’ comments:

As major revision I make three considerations

Reviewers’ comments: The methodology applied to respond to the objectives is correct, however, from my point of view, additional statistical analyzes are required for a better response. Why do the authors perform a Spearman correlation? In abstract section authors indicate Pearson Correlation. 

Authors’ comments: Thank you very much for the correction, it has been corrected, indicating in the abstract that it is Spearman's correlation.

Reviewers’ comments: Furthermore, I propose to the authors to carry out two additional tests: Multiple stepwise regression analysis, with the aim of identifying the most related component of physical condition with dependent variables.

Authors’ comments: Thanks for the proposal, this analysis has been carried out and added to table 6. Likewise, this same analysis has been written in the results with the 3 strength tests in a differentiated way as we did not know whether it was this analysis that the reviewer was referring to or the one we have added to table 6.

Reviewers’ comments: One-way ANOVA to analyze whether there are differences in physical condition between subjects with differential points in victimisation and aggression in bullying.

Authors’ comments: In the early years of the study of bullying, the questionnaires were analysed by transforming the subjects according to the role they assumed in bullying (victimizer, aggressor, aggressor-victimized...). Later, with the development of scales with greater psychometric properties, such as the one used in the study, victimization and aggression began to be studied quantitatively, as we have done in this study. For this reason, we have not divided the subjects into roles. Therefore, as we worked with quantitative data and not in groups of e.g. victims and bullies, we worked with such an analysis and could not work with the continuous dimensions in the One-way ANOVA.

Reviewers’ comments: Second, while the study appears to be sound, however, the authors should rewrite their Introduction and Discussion. Introduction: Since previous studies have analyzed the object of study of this manuscript, I consider it necessary to include quantitative results of these studies in the introduction section. 

Authors’ comments: Thank you very much for your appreciation, we have added information on the quantitative results of the studies most closely related to ours.

Reviewers’ comments: Third: what is the difference with Garcia-Hermoso A, Oriol-Granado X, Correa-Bautista JE, Ramírez-Vélez R. Association between bullying victimization and physical fitness among children and adolescents. Int J Clin Health Psychol. 2019 May;19(2):134-140. doi: 10.1016/j.ijchp.2019.02.006. Epub 2019 Apr 17. PMID: 31193131; PMCID: PMC6517651.

Authors’ comments: This study cited by the reviewer is a breakthrough in the study of the subject. The main differences with respect to ours is that we analyse not only victimisation in bullying but also aggressiveness. In addition, a greater number of muscle strength tests are carried out, in this study only manual dynamometry is used, while we also use the horizontal jump and the 30-second sit-up test. The analysis we carried out in our study is quantitative and therefore provides more information, whereas in the study of our colleagues we carried out a role-based analysis. Likewise, our study delves into the analysis of the behaviours of victimisation and aggression in the ordinal regression analysis.

Reviewers’ comments: The authors should revise the language to improve readability.

As minor revisions 

First paragraph, no relationship between first and second sentence is shown.

Line 24: delete "n"

Line 31: Pearson or Spearmar?

Line 36-40: Conclusions: review subsequent comment regarding conclusions

Line 47: delete “,”

Line 80-83: Can authors justify this argument with a reference?

Line 93: delete " [".

Line 100: “…these focusing on the role of the victim and concluding that good physical fitness may…” ¿Good or high physical fitness? At this point, it would be interesting for readers that authors reported more information.

Line 105-107: Idem, provide more specific contents

Authors’ comments: This information has been added to correct the previous minor changes.

Reviewers’ comments: 

Line 118-124: The authors say that: “…descriptive study involved 1035 schoolchildren aged 12 to 19 years…” However, later it is indicated that they are second year of baccalaureate and that those over 17 years were excluded. Clarify

Authors’ comments: Thank you very much for the detailed review, these issues have been corrected.

Reviewers’ comments: Materials and methods section. 

I propose to the authors a modification of the distribution of this section for a better understanding. Include a section on design or experimental design. Below is the description of the participants and later the procedure and statistical analysis.

Line 150: Why did the authors carry out the analysis of body composition with Tanita? What was the protocol used for a correct measurement using es? Where are the results provided?

Authors’ comments: The Tanita scale was used to measure the weight of the subjects but for logistical reasons it was not possible to carry out the analysis of body composition.

Reviewers comments: Line 153: m2

Line 155: substitute “training” for “familiarization”.

Line 156: Three? 

Line 188: The authors use Course Navette and Multistage Shuttle Run Test interchangeably, taking into account the original reference to use Multistage Shuttle Run Test. In addition, this test estimate the cardiorespiratory fitness or aerobic fitness not cardiovascular endurance

Line 210-211 I agree with the authors: Given the cross-sectional nature of this study, this prediction can only be interpreted on a theoretical basis and no causal relationship can be established. Therefore, the conclusions have to be reformulated.

Authors’ comments: Thank you very much for the detailed review, the issues have been corrected.

Reviewers’ comments: Results section.

Table 2: I propose to align the content to the left. Replace “,” for “.” to indicate decimals. Indicate the units in all variables

Table 3: Idem

Authors’ comments: Thank you very much for the detailed review, the issues have been corrected.

Reviewers’ comments: Line 239: ¿ weight or BMI?

Authors’ comments: It has been corrected by putting BMI.

Reviewers’ comments: Table 4: Indicate the units in all variables. Confidence intervals can be given in Table 4

Authors’ comments: Thank you very much for the suggestion, it has been added.

Reviewers’ comments: Line 248-250: Can the authors confirm that this sentence is correct?

Authors’ comments: Yes, it is correct it refers to the outcome of the complete victimisation model and the aggression model.

Reviewers’ comments: Line 252: “… being higher in boys…” How much?

Authors’ comments: This is shown in table 5 with β=.084 and t=2.201. If necessary it can be rewritten in the results but I think it is excessive if it is shown in the table.

Reviewers’ comments: Line 252: Is correct to age?

Authors’ comments: Thank you very much for your appreciation it has been deleted.

Reviewers’ comments: In Tables 6 and 7 the authors report the ordinary regression for some questions. Did the rest of the questions show no relationship? This is important for the correct writing of the conclusions and results.

Authors’ comments: This issue has been clarified in conjunction with the following commentary by rewriting these paragraphs.

Reviewers’ comments: Line 257-258: In this paragraph, results of table 6 are presented. The results are not related to the table results. Authors can rewrite this paragraph for better understanding. Also, it is not necessary to duplicate the results (text and table)

Authors’ comments: This issue has been clarified in conjunction with the above commentary by rewriting these paragraphs.

Reviewers’ comments: Discussion section.

The discussion section has to be rewritten. The authors abuse a chaining of paragraphs of results without arguing the same. The structure “In relation to...” is reiterative. It is necessary to discuss providing data from previous studies and comparing with those of this research, not just describing.

Authors’ comments: Thank you very much for your suggestions, the discussion has been rewritten with the authors' suggestions in mind.

Reviewers’ comments: Conclusion

Do the results support the conclusions? The authors have to rewrite the conclusions, contributing only those that correspond to the results obtained. Furthermore, a correlation is not a cause-effect relationship, so it must be rewritten.

Authors’ comments: Thank you very much for the correction, this paragraph has been rewritten.

Reviewers’ comments: Is the study free of limitations?

Authors’ comments: This information has been completed in the relevant section.

Reviewers’ comments: - It would be interesting to have more data regarding the sample and the data collection procedure. Has any participant had to be ruled out for medical reasons? Have you ever wondered if the participants have any pathology that could interfere with their performance?

Authors’ comments: This information has been added.

Reviewers’ comments: - There is no data on how the fitness tests were carried out in the sample (if, for example, it was part of the physical education class or if the measurements were made in the same time slot)

Authors’ comments: This information has been added.

Reviewers’ comments: - Finally, indicate that the writing of the manuscript should be revised to improve its reading: the theoretical introduction presents short sentences that can be joined (for example, line 77, line 250 R2 should be corrected)

Authors’ comments: Thank you very much for your comments and the suggestions have been corrected.

Editor comments: 3. Thank you for stating the following in the Acknowledgments Section of your manuscript:

“This project was funded by the Directorate General for Innovation and Teacher Training of the Counseling of Education of Andalusia (PIV-021/20) as an educational research project to be implemented in 2020.”

Authors’ comments: We have deleted the information in the text.

“This project was funded by the Directorate General for Innovation and Teacher Training of the Counseling of Education of Andalusia (PIV-021/20) as an educational research project to be implemented in 2020.”

Authors’ comments: This is how it should appear in the relevant section.

 “This project was funded by the Directorate General for Innovation and Teacher Training of the Counseling of Education of Andalusia (PIV-021/20) as an educational research project to be implemented in 2020.”

4. We suggest you thoroughly copyedit your manuscript for language usage, spelling, and grammar. If you do not know anyone who can help you do this, you may wish to consider employing a professional scientific editing service.

Authors’ comments: The professional person who has been in charge of reviewing the article in English is: Inmaculada C. Martí Garrido. Qualification: Degree in English Philology. Company: Educo Center Córdoba

I hope our work is satisfactory and we thank the editor and reviewers for their hard work in reviewing it.

Best regards

Juan de Dios Benítez Sillero

Universidad de Córdoba

---

## [Editor Report · Decision Letter 1]

13 Oct 2021

Behaviours involved in the role of victim and aggressor in bullying: relationship with physical fitness in adolescents.

PONE-D-21-09225R1

Dear Dr. Benítez-Sillero,

We’re pleased to inform you that your manuscript has been judged scientifically suitable for publication and will be formally accepted for publication once it meets all outstanding technical requirements.

Kind regards,

Celestino Rodríguez, PhD

Academic Editor

PLOS ONE
---

## [Editor Report · Acceptance letter]

19 Oct 2021

PONE-D-21-09225R1 

Behaviours involved in the role of victim and aggressor in bullying: relationship with physical fitness in adolescents. 

Dear Dr. Benítez- Sillero:

I'm pleased to inform you that your manuscript has been deemed suitable for publication in PLOS ONE. Congratulations! Your manuscript is now with our production department. 

Kind regards, 

on behalf of

Dr. Celestino Rodríguez 

Academic Editor

PLOS ONE